# Age-Related Mucus Barrier Dysfunction in Mice Is Related to the Changes in Muc2 Mucin in the Colon

**DOI:** 10.3390/nu15081830

**Published:** 2023-04-11

**Authors:** Xueqin Sang, Qingyu Wang, Yueyan Ning, Huihui Wang, Rui Zhang, Yixuan Li, Bing Fang, Cong Lv, Yan Zhang, Xiaoyu Wang, Fazheng Ren

**Affiliations:** 1College of Food Science and Engineering, Gansu Agricultural University, Lanzhou 730070, China; 2The Key Laboratory of Geriatrics, Beijing Institute of Geriatrics, Institute of Geriatric Medicine, Chinese Academy of Medical Sciences, Beijing Hospital/National Center of Gerontology of National Health Commission, Beijing 100730, China; 3State Key Laboratories for Agrobiotechnology, College of Biological Sciences, China Agricultural University, Beijing 100083, China; 4Key Laboratory of Precision Nutrition and Food Quality, Department of Nutrition and Health, China Agricultural University, Beijing 100193, China; 5Key Laboratory of Functional Dairy, Co-Constructed by Ministry of Education and Beijing Municipality, College of Food Science & Nutritional Engineering, China Agricultural University, Beijing 100083, China; 6Food Laboratory of Zhongyuan, Luohe 462000, China

**Keywords:** aging, mucus, microbiota, glycosyltransferase, goblet cells

## Abstract

During aging, the protective function of mucus barrier is significantly reduced among which changes in colonic mucus barrier function received the most attention. Additionally, the incidence of colon-related diseases increases significantly in adulthood, posing a threat to the health of the elderly. However, the specific changes in colonic mucus barrier with aging and the underlying mechanisms have not been fully elucidated. To understand the effects of aging on the colonic mucus barrier, changes in the colonic mucus layer were evaluated in mice aged 2, 12, 18, and 24 months. Microbial invasion, thickness, and structure of colonic mucus in mice at different months of age were analyzed by in situ hybridization fluorescence staining, AB/PAS staining, and cryo-scanning electron microscopy. Results showed that the aged colon exhibited intestinal mucus barrier dys-function and altered mucus properties. During aging, microorganisms invaded the mucus layer to reach epithelial cells. Compared with young mice, the thickness of mucus layer in aged mice in-creased by 11.66 μm. And the contents of the main components and glycosylation structure of colon changed. Among them, the proportion of goblet cells decreased significantly in older mice, and the expression of spdef genes that regulate goblet cell differentiation decreased. Further, the expression of key enzymes involved in mucin core structure formation and glycan modification also changed with aging. The expression of core 1 β1,3-galactosyltransferase (C1GalT1) which is the key enzyme forming the main core structure increased by one time, while core 2 β1,6 N-acetylglucosaminyltransferase (C2GnT) and core 3 β1,3 N-acetylglucosaminyltransferase (C3GnT) decreased 2 to 6- and 2-fold, respectively. Also, the expression of sialyltransferase, one of the mucin-glycan modifying enzymes, was decreased by 1-fold. Overall, our results indicate that the goblet cells/glycosyltransferase/O-glycan axis plays an important role in maintaining the physicochemical properties of colonic mucus and the stability of intestinal environment.

## 1. Introduction

In recent years, the average human life expectancy has increased significantly. The 2019 revision of the United Nations’ World Population Prospects predicts that the proportion of people aged 65 years and older will increase by almost 16% by 2050 [1]. As the population ages, the incidence of aging-related chronic diseases has increased year by year, among which the incidence of gastrointestinal chronic diseases has increased significantly [2]. Age-related changes in intestinal physiological function could lead to various gastrointestinal diseases including, polyps, inflammatory bowel dis-ease, and so on [3]. Recently, the incidence of inflammatory bowel disease has significantly increased with age, but the exact pathogenesis remains unclear. Ulcerative colitis is a common inflammatory bowel disease that is mainly caused by deterioration of colonic mucus layer [4]. It has been reported that mucus layer becomes thinner and the inner mucus layer becomes penetrable during the development of colitis [5]. Overall, it could be concluded that intestinal dynamic homeostasis depends on the protective function of intestinal mucus barrier, and its disruption is one of the main causes of inflammation [6].

The mucosal barriers that maintain a normal dynamic balance of intestinal environment mainly include mucus barrier, intestinal epithelial barrier and immune barrier [7]. Mucus barrier is the main defensive layer that separates microbiota from epithelial cells. It consists of mucus secreted by goblet cells and overlies intestinal epithelial cells [8,9]. The small intestine has only one mucus layer, whereas the colon has two mucus layers [10], an outer layer that allows bacterial colonization and an inner layer that does not allow luminal bacteria to penetrate [11]. Thus, mucus layer is not only a protector of intestinal epithelial cells but also a key mediator of host-bacterial interactions [12]. And the colonic mucus barrier is a major defender of intestinal pathogens [13]. The physiological importance of mucus layer has been demonstrated over the years by studying patients with ulcerative colitis. These studies reported several defects in mucus layer function, including a decrease in mucus layer thickness and mucus chemical composition, and an increase in bacterial penetration of the mucus barrier [12]. Similar results were obtained in the literature related to changes in the mucus layer during aging in mice. It was speculated that such changes were related to alterations in mucus composition, but the mechanism of interaction between mucus composition and mucus functional defects has not been elucidated yet [14]. 

Muc2 mucin is a major component of the mucus layer [15]. The colonic mucus layer formed by Muc2 mucin builds a functional barrier between the host and bacterial microbiota [16]. Studies have shown that Muc2-deficient mice spontaneously developed inflammation and were more susceptible to infection by pathogens [17]. Furthermore, Muc2 is a high molecular weight glycoprotein formed by glycosyltransferases. Mucin glycosylation is mainly responsible for the formation of four key O-glycan core structures [18]. The major polysaccharides in colonic mucus layer are mucin O-glycans derived from core 1 and core 3 [19]. The key enzymes involved in the formation of core 1 and core 3 O-glycan chains are core 1 β1,3-galactosyltransferase (C1GalT1) and core 3 β1,3 N-acetylglucosaminyltransferase (C3GnT), respectively [20,21,22]. Core 1 O-glycans form core 2 O-glycan with the participation of core 2 β1,6 N-acetylglucosaminyltransferases (C2GnT) [18]. C2GnT1 and C2GnT3 are normally expressed in intestinal epithelial cells, and C2GnT2 is also known as mucus-type C2GnT2 because it is highly restricted to mucus-producing tissues which highly ex-presses in stomach and colon. Studies using a C2GnT2-deficient mouse model found that C2GnT2 deficiency impaired the mucosal barrier and increased mucosal barrier permeability and susceptibility to colitis. Previous studies using C1galt1 and C3GnT deficient mouse models (IEC C1galt1-/-, C3GnT-/- mice) have demonstrated that mucin-type O-glycans are key factors in protecting the intestinal mucus barrier from damage [20,22]. It has also been shown that defect in O-glycan chains lead to alterations in the composition of intestinal flora, resulting in bacteria-dependent intestinal inflammation [23]. Besides, recent studies have reported distal colonic O-glycosylation impaired during aging, and protein expression of the rate-limiting enzyme in O-glycosylation, T-synthetase, decreased with age [1]. However, mechanisms under-lying the interaction between mucin content, mucin glycosylation structure and mucus layer function during aging need to be further investigated.

Here, naturally aged mice were used to investigate how the physiological, physical and chemical properties of mucus altered during aging. This study focused on compositional and structural changes in mucus during aging and the underlying mechanisms.

## 2. Materials and Methods

### 2.1. Animals

Male C57BL/6J mice used in the experiment were purchased from the Laboratory Animal Center of Weitong Lihua Laboratory Animal Technology Company (Beijing, China). Mice of the same age (months) were purchased in chronological order of 2, 12, 18, and 24 months of age. Mice of each month of age (*n* = 12) were placed in individually ventilated cages with ad libitum access to mouse growth and reproduction diet (12.95% fat) (Beijing Keao Xieli Feed Co., Ltd., Beijing, China) and autoclave-sterilized water. All mice were sacrificed using appropriate anesthesia and/or cervical dislocation before sample collection. The experiments and protocols were approved by the Ethics Committee of China Agricultural University (AW50102202-5).

### 2.2. Mucus and Tissue Collection

Colon tissue was incised longitudinally and the mucus was gently scraped with a smooth slide after careful removal of the fecal contents, and the mucus samples were frozen rapidly in liquid nitrogen. In addition, colon tissues were immediately frozen in liquid nitrogen and then stored at −80 °C for subsequent RNA and protein assays.

### 2.3. pH Measurements

The colon was cut longitudinally, and mucus was gently scraped from the entire colon with a slide after the feces were gently removed. About 100 to 120 μL of colonic mucus was scraped from each mouse, and pH was measured using a portable pen acidity meter (Yanlin Laboratories (Shenzhen) Co., Shenzhen, China). The samples were immediately frozen in liquid nitrogen after the measurements and stored at −80 °C for further experiments [24].

### 2.4. Water Content and Mucus Weight Determination 

This method is modified based on methods from the literature. Briefly, the collected mucus samples were quickly placed into a freeze-dryer at a cold trap temperature of −50 °C for 48 h [24]. At the end of freeze-drying, the samples were placed in a desiccator for 30 min and weighed. The weight per unit length of mucus was calculated by the weight of collected mucus divided by the length of collected colonic tissue [24].

### 2.5. In Situ Hybridization Fluorescent Staining

The methods refer to the literature and are briefly described as follows [1,19,25]. Colon sections were stained with universal bacterial probe EUB338 (*50-GCTGCCTCCCGTAGGAGT-30*; bp 337–354 in bacteria EU622773) at 50 °C overnight. A nonspecific probe (*NON338 50-ACTCCTACGGGAGGCAGC-30*) was used as negative control. For Muc2-Fish double labeling experiments, Muc2 immunofluorescence staining was performed on sample sections before Fish labeling. Sample sections were incubated with anti-Muc2 (Service bio, GB11344, Wuhan Sevier Biotechnology Co., Ltd., Wuhan, China, incubated at 4 °C for 12 h) and goat anti-mouse IgG (H + L) cross-adsorbed secondary antibody, Alexa fluor™ 488 (Invitrogen, A-11001, Thermo Fisher Scientific, Beijing, China, incubated at room temperature for 1 h).

### 2.6. Cryo-Scanning Electron Microscopy and Image Analysis

The preparation of samples, Cryo-SEM, and image analysis were taken from the literature and modified [26]. In brief, samples were thawed at room temperature, and a single drop of each sample was cast onto a metal holder and then frozen in liquid nitrogen. Frozen samples were transferred into the sample preparation bin for freeze-fracture and sublimated under vacuum for 15 min at −80 °C. The Cryo-SEM images were counted for hole size and shape using ImageJ software, with at least 100 holes involved in each field of view. The minimum diameter of the FERET, which refers to the shortest distance between any two parallel tangents of the hole, was used to describe the size of the hole. The Aspect Ratio (AR) was used to describe the pore shape, which was described as the ratio of the largest ferret diameter to the smallest ferret diameter.

### 2.7. Immunostaining

Colon tissue was collected and the mesentery was gently removed with forceps. Then, the tissue was ligated at both ends with surgical thread and fixed in Carnoy’s fixative solution for no more than 12 h. After fixation, the tissues were dehydrated and embedded in paraffin. Mucus thickness staining was performed using the AB-PAS Stain Kit (Solarbio, G1285, Beijing Solarbio Science & Technology Co., Ltd., Beijing, China) following manufacturer instructions [27]. PS software (Adobe Photoshop 2021 22.0.0, Adobe Systems Incorporated, San Jose, CA, USA) was used to measure the thickness of the mucus layer [19,28].

For IHC, the sections were subjected to antigen retrieval in 0.01 M sodium citrate for 20 min. The nonspecific antibody binding was blocked by goat serum working solution (Zhongshan Jinqiao, SP-9002, Beijing ZhongShan JinQiao Biotechnology Co., Beijing, China) for 1 h. Following blocking, rabbit anti-Muc2 (Service bio, GB11344, Wuhan Sevier Biotechnology Co., Ltd., Wuhan, China) antibody was used to incubate the sections at 4 °C overnight. After 30 min, sections were stained with the corresponding immune secondary antibody for 30 min. Finally, the slices were stained with DAB chromogenic kit (ZSGB-BIO, ZLI-9019, Beijing ZhongShan JinQiao Biotechnology Co., Beijing, China), counterstained stained with hematoxylin, dehydrated, and sealed.

For immunofluorescence [29], colon tissue sections (4 μm) were dehydrated, antigen-repaired, and blocked for 1 h in blocking reagent at room temperature. Staining was performed with rabbit anti-Muc2 (Service bio, GB11344, Wuhan Sevier Biotechnology Co., Ltd., Wuhan, China) antibody and placed at 4 °C overnight. Primary antibody identification was performed with a fluorescent secondary antibody and incubated at room temperature for 1 h. Finally, the nuclei were stained with DAPI (Beyotime Biotechnology, A-11008, Shanghai Biyuntian Biotechnology Co., Shanghai, China) for 10 min.

### 2.8. Real-Time PCR

To determine the effect of age on gene expression, total RNA was extracted from colon tissues using a specialized kit (Qiagen, Germantown, MD, USA) [3]. The RNA was then reverse transcribed into cDNA using a reverse transcription kit (abm, G592, Zhenjiang Ebimont Biotechnology Co., Jiangsu, China). The mRNA expression of O-glycosylase (C1galt1, C2gnt, C3gnt, and St6galnac-I) and goblet cell precursor markers (spdef, klf4, Beijing Liuhe Huada Gene Technology Co., Beijing, China, and muc2) was quantified on a real-time PCR system using the primers shown in Table 1.

### 2.9. Statistical Analysis

Data were expressed as mean ± SD. GraphPad Prism 8.0.2, (GraphPad Software, Inc. San Diego, CA, USA) was used for all statistical analyses. All samples had biological replicates. Data were statistically analyzed using one-way ANOVA with Tukey’s post hoc test [30]. 

## 3. Results

### 3.1. Mucus Barrier Function

The mucus layer is the first protective layer of mucosa overlying the intestinal epithelial cells, and its integrity is essential for the maintenance of health [31]. As shown in Figure 1A, colonic mucus layers at 2, 12, and 18 months of age were intact and not invaded by microorganisms. In contrast, the mucus layer of 24-month-old mice was more loosely structured and colonized by a certain number of microorganisms. To investigate the effect of age on mucus thickness, AB-PAS staining on colonic tissue samples was performed. Surprisingly, results showed a continuous increase in mucus layer thickness in aged mice (Figure 1B,C). The thickness of the mucus layer in mice aged 12 to 24 months was significantly higher than that in mice aged 2 months. These results indicate that the function and thickness of the mucus layer are affected during aging, and the mucus barrier deteriorates with aging.

### 3.2. Microscopic Characterization

The disulfide bonds between non-glycosylated region and multiple non-covalent bonds form a highly entangled mucus gel network. These networks are important structural components in the mucus layer that can interact with microbes and protect intestinal epithelial cells. The Cryo-SEM results of different ages were displayed in Figure 2A–C. Results showed that there were extended gel networks in the colonic mucus samples of all mice, and the size of the pores of these gel networks was statistically different in different age groups (Figure 2A,B). However, the shape of the pores did not change significantly (Figure 2C). Cryo-SEM images of the mucus at 2 months of age showed relatively large pores, while the colonic mucus network at 12, 18, and 24 months of age contained smaller pores. Compared with 2-month-old mice, the mini-mum diameter of the colonic mucus network was significantly reduced in 12-month-old, 18-month-old, and 24-month-old mice, especially at 12 months old. This suggests that the pore size of the mucus microscopic network changes during aging, but the network shape does not.

### 3.3. Physiological Properties

pH is one of the influencing factors in the formation and release of mucus [32,33]. The changes in mucus pH with increasing age were shown in Figure 3A. Compared with 2 months of age, pH did not change significantly at 12 months of age while decreased significantly at 18 and 24 months of age. Changes in mucus pH indicate that mucus secretion and stratification may also be altered.

The results of mucus weight and water content are shown in Figure 3B,C. The mucus weight per unit length of mice at 12, 18, and 24 months was significantly increased compared with that of mice at 2 months. The average water content of colonic mucus in 2-month-old mice was 79.12%. Compared with 2 months of age, there was a significant decrease at 12, 18, and 24 months of age. In contrast, the trend of mucus dry matter content was opposite to that of water content, with a significant increase at 24 months of age compared with 2 months of age (Figure 3D). These results indicate that the mucus layer becomes more viscous due to decreased water content and increased dry matter content during aging.

### 3.4. Muc2 Mucin

The gel-forming mucin Muc2 is a major structural and functional component of the mucus barrier [34]. Immunofluorescence staining of colon sections at different months of age was performed, and Muc2 gene expression in the colon was quantified. Compared with mice at 2 months of age, Muc2 protein content in the colon decreased significantly in mice at 24 months of age (Figure 4A,B). Interestingly, this decrease occurred mainly in the crypt base, which may have an impact on the activity of stem cells. Similarly, the Muc2 mRNA level reduced in 24-month-old mice compared with 2-month-old mice (Figure 4C).

### 3.5. Goblet Cells and Genes Regulating Goblet Cell Formation

Goblet cells are a cell type specializing in secreting mucus [35]. The number and proportion of goblet cells are shown in Figure 5A–C. The average number of goblet cells in the colon tissue of mice aged 2, 12, 18, and 24 months was 12, 14, 14, and 12, respectively (Figure 5B). Results showed that the number of goblet cells in aging mice did not change significantly compared with young mice, but the proportion of goblet cells decreased significantly at 24 months of age (29%) (Figure 5C), suggesting that the ability of goblet cells secreting mucus was inhibited with age. 

Spdef is a goblet cell marker gene, which regulates the differentiation of intestinal goblet cells. Klf4 is also involved in the terminal differentiation of goblet cells. Compared with 2-month-old mice, the mRNA level of spdef was significantly reduced in 12-month-old, 18-month-old, and 24-month-old mice (Figure 5D). In contrast, the expression of klf4 was significantly higher in 12-, 18-, and 24-month-old mice than in 2-month-old mice (Figure 5E). 

### 3.6. Mucin Glycosylation

Core enzymes are known to be directly involved in mucin glycosylation, forming important core structures. C1GalT1 and C1GalT2 are key enzymes in core 1 O-glycan biosynthesis and are widely expressed in epithelial cells. Compared with 2-month-old mice, the expression of C1galt1 in the colon of 12-, 18- and 24-month-old mice significantly increased, but there was no significant change in the expression of C1galt1 in the colon of 18- and 24-month-old mice (Figure 6A). The expression of C1galt2 was shown in Figure 6B. Compared with 2-month-old mice, the expression of C1galt2 was significantly increased in the colon of 12-month-old mice, while no significant changes were observed in the colon of 18-month-old and 24-month-old mice. The core 2 O-glycan is an important component of Muc2 O-glycans which is regulated by the core 2 transferase family (C2gnts). Compared with 2-month-old mice, C2gnt1 expression was significantly increased in the colon of 12- and 18-month-old mice and decreased in the colon of 24-month-old mice (Figure 6C). Meanwhile, the expression of both C2gnt2 and C2gnt3 decreased significantly with age compared with 2 months old mice (Figure 6D,E). C3gnt is the only enzyme catalyzing the biosynthesis of core 3-derived O-glycan chains [22]. Results showed that the expression of C3gnt in mice at 12 and 24 months was significantly reduced compared with that of mice at 2 months, and the expression of mice at 24 months was about half that of mice at 2 months (Figure 6F).

ST6GALNAC1 is a major sialyltransferase that specifically expresses in goblet cells and is sialylated at the ends of glycans on intestinal mucus [36]. Compared with younger mice, the expression of St6galnac-I in colon tissues of mice aged 12, 18, and 24 months was significantly decreased, and the expression level of St6galnac-I in mice aged 18 and 24 months was about 50% of that of mice aged 2 months (Figure 6G). Compared with 2-month-old mice, the expression of St6galnac-II was significantly increased in 18-month-old mice, and decreased in 12-month-old and 24-month-old mice (Figure 6H).

## 4. Discussion

Gut is one of the organs that undergoes significant changes during the aging process and the incidence of gut-related diseases in the elderly is quite high [37]. Mucus layer is one of the intestinal barriers, which prevents microorganisms from directly contacting intestinal epithelial cells to maintain host health [29]. The protective function of mucus barrier mainly depends on the mutual relationship between commensal microorganisms and mucus layer [38]. Our study showed that the mucus layer of 2-month-old mice was intact and free of microorganisms, whereas 24-month-old mice were invaded by bacteria. This observation is consistent with a previous study in which the aged (24 months) colonic mucus layer allowed bacterial penetration, but the inner mucus layer was intact in 2-month-old and 16-month-old mice [1]. This implies that with aging the protective function of mucus barrier is compromised and cannot prevent harmful microorganisms from reaching intestinal epithelial cells. Interestingly, the thickness of mucus layer increased during aging, which is different from previous studies [28]. In the present study, we found that the thickness of the colonic mucus layer increased at 12 and 24 months of age, whereas at 18 months of age, it increased compared with 2 months of age but significantly decreased compared with 12 and 24 months of age. A decreasing trend of mucus thickness was observed at 18 months compared with 12 months, which may be due to the beginning of aging in mice starting from 18 months of age. Gut may respond to changes in the body with a transient decrease in the thickness of the mucus layer. The thickness of mucus at 24 months of age was significantly higher than that at 18 months of age, which may be due to the slow peristalsis of the intestine in old mice and the slow mucus elimination. However, mucus thickness was not necessarily a useful indicator of mucus barrier function. Previous studies have shown that all mouse models of colitis had bacteria in contact with epithelial cells, but IL-10^−/−^ mice showed a thicker mucus layer than wild-type mice [39]. Based on the results of microbial invasion into the colonic mucus layer and the increased mucus thickness, we hypothesized that this might be due to the changes in mucus composition and mucin structure during aging. Although thickness of the mucus layer increases during aging, it is also possible that changes in mucus composition make the mucus layer easier for microbial attachment, and that changes in mucin structure make the mucus layer loose and easier for microbial invasion.

As mentioned earlier, the major component of intestinal mucus layer is gel-forming mucin, which acts together with other components to protect and lubricate GI tract [40]. The species of mucins varied in different GI segments. Muc2 is essential for colon protection, and Muc2-deficient mice spontaneously develop colitis [17]. It has been reported that aged mice (19 months) exhibited impaired mucus barrier without colitis. However, Muc2^−/−^ mice lacking the secretory mucus layer developed spontaneous colitis around 6 weeks of age. This further suggests that muc2 plays an important role in the colonic mucus layer [28]. The importance of muc2 in the mucus of aged mice was also demonstrated by experiments in a metformin-treated model which found that metformin reduced intestinal leakage by increasing Muc2 expression in the colon of aged mice. In our study, it was also found that Muc2 protein content decreased at 24 months of age. However, Muc2 gene expression increased at 12 months of age but did not differ significantly at 2, 18, and 24 months of age. The reason for this inconsistent may be that Muc2 gene is subject to many different regulations during transcription and translation, and these transcriptional, post-transcriptional, and translational regulations all play roles in protein expression. Since goblet cells are major sources of Muc2, the changes in goblet cell number and proportion with age were further examined. There was no change in the number of colonic goblet cells but a reduction in proportion. This result is consistent with a previous study showing that mucin expression decreased in aged mice (24 months) but the number of goblet cells in colon did not change [1]. These results indicate that the ability of goblet cells secreting mucin decreases during aging. Studies using Muc2 deficient mouse models have re-ported that goblet cells appear to be absent in the entire colon of muc2^−/−^ mice and have smaller goblet cell morphology compared to muc2^+/+^ mice [17]. This further suggests that goblet cells play a crucial role in mucus secretion. To explore the mechanism by which goblet cells are affected during aging, the expression of genes regulating goblet cell formation was determined. It was found that the expression of gene spdef was significantly decreased with aging which is consistent with previous studies [3]. These results indicate that the expression of regulatory factors controlling goblet cell formation is inhibited during aging, which prevents the formation of goblet cells, thereby reducing mucin secretion and leading to weakened mucus barrier function.

In addition to changes in the content of the main components of mucus, changes in the microstructure of mucus are also associated with mucus barrier dysfunction. Mucins are divided into two major subfamilies, including transmembrane mucins and gel-forming mucins [41]. The monomer structure of gel-forming mucin Muc2 has been extensively studied, and the monomer creates a highly entwined gel network through disulfide bonds [42]. Network structures were observed in all cryo-electron microscopy images of colonic mucus in this study. Results showed that the pore size of the network structure decreased with aging, and the pore shape was round or nearly round in all ages of mice. The same shape of colonic mucus was also reported in another study in which mucus microstructure was examined in different segments of porcine GI tract. At the same time, it is also speculated that the size of the network structure may be related to bacteria, and small pores can hinder the diffusion of bacteria without impeding the diffusion of other substances [26]. The results of bacterial invasion during aging could also be explained by the hypothesis that mucus blocks bacterial invasion by changing the size of the network during aging. In this study, we also found that the minimum pore size of the colonic mucus network was significantly reduced at 12 months of age compared with 18 and 24 months of age. The microstructure of the colonic mucus network has been reported to be dependent on factors other than mucus type. It has been established that mucus networks are formed by the supramolecular assemblies and that mucin gel assemblies is governed by the collective action of non-mucins, disulfide bridges, Ca^2+^-mediated links and hydrogen bonds, so that disruption of any single element results in alterations of the network [42]. 

Based on the results of mucus microstructure changes, the underlying mechanism was further investigated. It was found that the change in the microstructure of mucus may be related to the glycosylation of mucin-type O-glycan. Several studies in mouse models lacking core 1 or core 3-derived O-glycans have found disruption of mucus layer and have shown that O-polysaccharides stabilize the structure of mucus layer [20]. The biosynthesis of core structure O-glycan is regulated by different transferases including core enzymes and modifying enzymes [43]. Our study found an increase in core-1 glycosyltransferase expression and a decrease in core-2 and core-3 transferase expression during aging. A previous study in which immature rats fed polyamines found that polyamines were involved in the regulation of mucin glycosylation during postnatal intestinal development by promoting the synthesis of glycoprotein galactosyltransferase [44]. This may explain the increased expression of galactosyltransferase during aging. We hypothesized that intestinal microbiota-derived polyamines increased during aging, thus leading to increased mucin galactosylation. Integrity of the core structure plays an important role in mucin molecules. Previous studies using mouse models of intestinal core 1- and core 3-derived O-glycans deficiency revealed that Muc2 mucin required both O-glycans to protect it from degradation by microbial-derived proteolytic enzymes [45]. Reduced expression of core 2 and core 3 transferases during aging makes it difficult to form core 2 and core 3 structures, thus truncating the O-glycan chains. Low expression of St6I and St6II transferase results in reduced mucin glycan modification. Short and less modified O-glycan chains may expose the protein core to proteases that degrade mucins. Overall, these results suggest that de-creased expression of mucin glycosylases during aging leads to truncation of O-glycans and degradation of protein core, which disrupts mucin microstructure and leads to decreased mucus barrier function.

## 5. Conclusions

In conclusion, this study explored the changes in the physical and chemical prop-erties of mucus layer with increasing age and the underlying mechanisms. We demon-strated that mucin content and structural changes in aging mice lead to impaired mucus barrier function. Furthermore, we found that low expression of genes regulating goblet cell formation reduced the percentage of goblet cells and mucin content, and low expression of mucin glycosylase resulted in shorter O-glycan chains, thereby affecting mucus microstructure with aging. Taken together, this study is important for better understanding the role of the intestinal mucus layer in the aging process and may provide potential targets for the treatment of intestinal diseases in the elderly.

## Figures and Tables

**Figure 1 nutrients-15-01830-f001:**
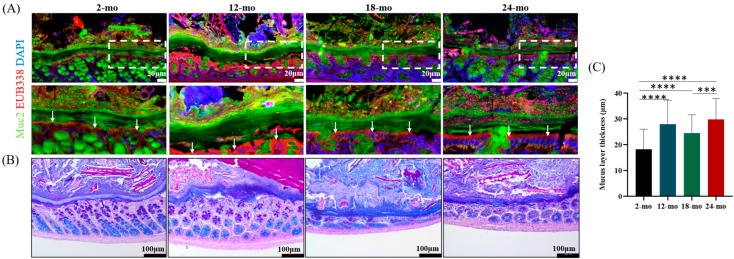
Aged colon showed impaired mucus barrier integrity and increased mucus thickness in mice. (**A**) FISH staining of distal colon sections using EUB338 probes and Muc2 antibodies showed bacterial penetration of the inner mucus layer. The magnified images showed the boxed region. White arrows indicated the apical surface of the epithelial cells. Scale bars: 20 μm. (**B**) AB/PAS staining images of the distal colons of mice aged 2, 12, 18, and 24 months. Scale bars: 100 μm. (**C**) Mucus thickness was measured in three colon tissues of mice aged 2, 12, 18, and 24 months (9 measurements per section, 3 sections per mouse). Data were presented as mean ± SD. Data were statistically analyzed using one-way ANOVA with Tukey’s post hoc test. Asterisk indicates significant difference, *** *p* < 0.001, **** *p* < 0.0001, *n* = 3 per group.

**Figure 2 nutrients-15-01830-f002:**
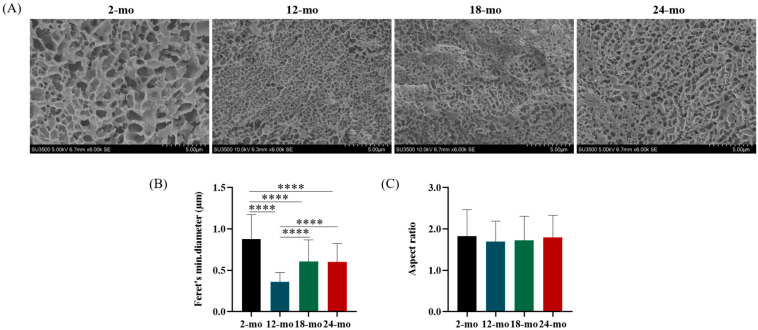
Microstructural characteristics of colonic mucus in mice. (**A**) Representative cryo-scanning electron micrographs of colon mucus at different ages (scale bars: 5 μm). (**B**,**C**) showed the Feret’s minimum diameter and aspect ratio distribution of mucus pores, respectively. At least 100 pores per micrograph were identified and used in the analysis. Data were presented as mean ± SD. Data were statistically analyzed using one-way ANOVA with Tukey’s post hoc test. Asterisk indicates significant difference, **** *p* < 0.0001, *n* = 3 per group.

**Figure 3 nutrients-15-01830-f003:**
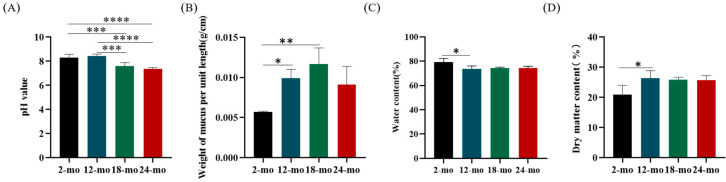
Physiological characteristics of colonic mucus in aging mice. The pH (**A**), mucus weight per unit length (**B**), water content (**C**), and dry matter content (**D**) of colonic mucus in aging mice were determined. Data were presented as mean ± SD. Data were statistically analyzed using one-way ANOVA with Tukey’s post hoc test. Asterisk indicates significant difference, * *p* < 0.05, ** *p* < 0.01, *** *p* < 0.001, **** *p* < 0.0001, *n* = 3 per group.

**Figure 4 nutrients-15-01830-f004:**
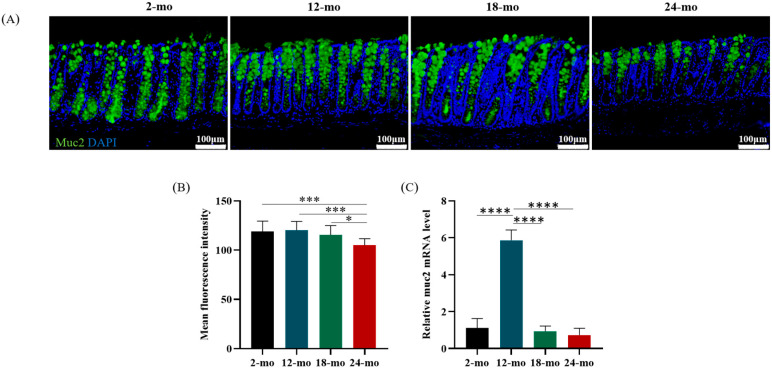
Muc2 mucin content of the mucus layer of colon in aging mice. (**A**) Immunofluorescence staining of 2-, 12-, 18- and 24-month-old mouse colon sections with anti-Muc2 (in green; cell nuclei in blue). Scale bars: 100 μm. Immunofluorescence results of Muc2 mucin (**B**) and relative mRNA expression (**C**) of Muc2 in the mouse colon. Data were presented as mean ± SD. Data were statistically analyzed using one-way ANOVA with Tukey’s post hoc test. Asterisk indicates significant difference, * *p* < 0.05, *** *p* < 0.001, **** *p* < 0.0001, *n* = 3 per group.

**Figure 5 nutrients-15-01830-f005:**
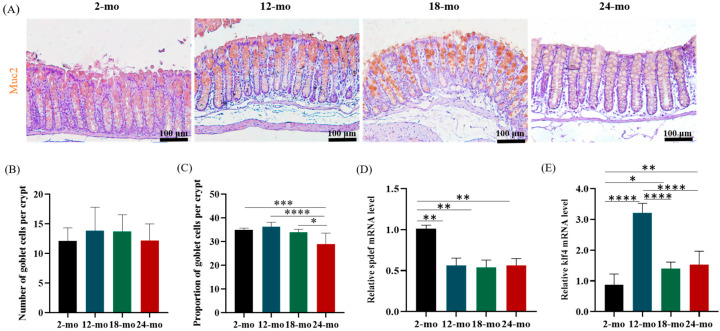
The number and proportion of goblet cells and the expression of genes regulating goblet cell formation changed in aging mice. Immunohistochemical staining of 2-, 12-, 18- and 24-month-old mouse colon sections with anti-Muc2 (in brown; cell nuclei in blue) (**A**), Scale bars: 100 μm. Number (**B**) and proportion (**C**) of goblet cells were measured. Relative mRNA expression of spdef (**D**) and klf4 (**E**) in the colon of mice. Data were presented as mean ± SD. Data were statistically analyzed using one-way ANOVA with Tukey’s post hoc test. Asterisk indicates significant difference, * *p* < 0.05, ** *p* < 0.01, *** *p* < 0.001, **** *p* < 0.0001, *n* = 3 per group.

**Figure 6 nutrients-15-01830-f006:**
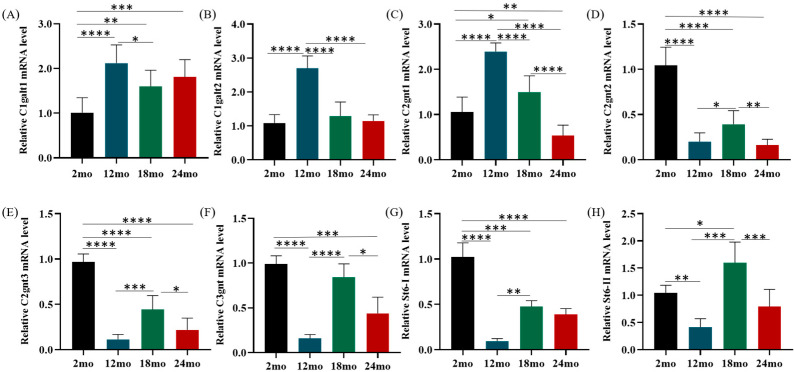
Altered mRNA expression of glycosylated transferase during aging. mRNA level of key enzymes in core 1 (**A**,**B**), core 2 (**C**–**E**), and core3 (**F**) formation in colon. Gene expression of the mucin-type O-glycan modifying enzymes (**G**,**H**) in colon tissue were measured. Data were presented as mean ± SD. Data were statistically analyzed using one-way ANOVA with Tukey’s post hoc test. Asterisk indicates significant difference, * *p* < 0.05, ** *p* < 0.01, *** *p* < 0.001, **** *p* < 0.0001, *n* = 3 per group.

**Table 1 nutrients-15-01830-t001:** Primer sequences of qRT-PCR.

Gene Name	Forward 5′-3′	Reverse 5′-3′
Muc2	GGGAATGTTGCAAGAAGTGC	TTTTGTGAATCTCCCCAGGC
Spdef	ACGTTGGATGAGCACTCG	CCATAAAAGCCACTTCTGCAC
Klf4	CGAACTCACACAGGCGAGAA	GAGCGGGCGAATTTCCA
C1galt1	TGGAATTACAACTATTATCCTCCCATA	CAACATAGTGAAAAGAAACTGCGATA
C1galt2	TGGAGCCGTTCTAGATGCGGAAAA	GGGGCTTGCAGATGGTGATGCT
C2gnt	GCAGCCAAGAAGGTACCAAA	ACAGGCGAGGACCATCAA
C2gnt1	GCTTGATAGGAACTTGGCAGCAC	CACCTTCTGGATTTCTTCTGGGTC
C2gnt2	ACCTTCACTCCACATCACTCACGG	TTATTCAGCAGAGCCTGGGTCACC
C2gnt3	GCCGCTGTTCTTGCTGTTTTG	AGTCACTTGTCATCGCCACGA
C3gnt	GGCCAGATTCTCCTCTCTCAAACG	AGTGCTCCGCTGTCCAGTCCA
St6galnac-I	TGTTAGGGACCAGCCATCCA	ATGAACTGGCACCTGGAATC
St6galnac-II	CGGATGTTGTTGCTCGTTGC	AGTCGGCTCTTTCTGTTTTCC
Gapdh	AGGTCGGTGTGAACGGATTTG	TGTAGACCATGTAGTTGAGGTCA

## Data Availability

Data will be sent upon request from the corresponding author.

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
