# Peer review of "Age-Related Mucus Barrier Dysfunction in Mice Is Related to the Changes in Muc2 Mucin in the Colon"

_nutrients, 2023, doi:10.3390/nu15081830_

Round 1

Reviewer 1 Report

In the manuscript, “The age-related mucus barrier dysfunction in mice is related to the changes of Muc2 mucin in colon” by Sang et al., the authors conduct a detailed analysis of age-related changes of the mucus and genes associated with mucus production in normal mice. They found a correlation with decreased epithelial barrier function and changes in the mucus components. Although, it is difficult to conclusively determine that these changes documented by the investigators are responsible for the decreases barrier function, the systematic documentation of mucus-associated genes and mucus architecture during aging should be of interest to researchers of mucosal immunity and gut functions. My comments are listed below.

1.     Major comment 1. Figure 1A. The infiltration of bacteria in the 24-month group is slightly difficult to detect even with the enlarged image. I recommend that the authors provide, if possible, an even higher resolution image of the 24-month FISH image of bacteria. Alternatively, use a larger sized image.

2.     Major comment 2. Figure 1A. It would be helpful to indicated (using arrows) the apical surface of the epithelium.

3.     Major comment 3. Figure 1C. The authors use “a”, “b”, “c”, and “d” to indicate statistically relevance. I strongly recommend that more conventional methods, such as asterisks. Also, please indicate the p values. Same applies to all figures showing statistically relevance.

4.     Major comment 4. Generally, 3 mice were used per group in most experiments. I am not a statistician but I feel that the numbers of mice per groups appears insufficient. Please re-evaluate the numbers of mice is necessary.

5.     Major comment 6. In some of the figures (example, Fig 6G, Figure 6H), the relative mRNA level of the 2-month sample is not set to one (1.0). I recommend that the 2-month mRNA data be set as 1.0. Please explain.

6.     Major comment 7. In the Materials and Methods section, please provide as much data as possible for the pH measurement of mucus. For example, was the entire mucus layer from 3 mice retrieved for pH measurement? What was the pH meter used for the measurement? What was the volume of the mucus sample? Etc.

7.     Major comment 8. The authors show that the although the mucus layer is thicker in the 24-month group compared the 2-month group but bacterial infiltration is increased. Please speculate in more detail possible explanations for this observation.

8.     Major comment 9. Please provide a rationale for using the 2, 12, 18, 24 month aged groups for this study.

9.     Minor comment 1. Figure 3, legend. Please move the figure 3 legend so that it is located immediately below the actual figure.

10.  Minor comment 2. Figure 4. Please shift the figure to the right for correct alignment. Other figures as well.

11.  Minor comment 3. Figure 5A. The figure lists DAPI was used in the IHC procedure. Please check of hematoxylin was used rather than DAPI.

12.  Minor comment 4. I recommend moving Table 1 to the materials and methods section or alternatively to a supplementary data section.

13.  Minor comment 5. Line 140. “antigen repair” should be “antigen retrieval”

14.  Minor comment 6. Line 147. “Immunofluorescence” should be “immunofluorescence”

Reviewer 2 Report

General comments

The manuscript of Xueqin Sang et al. reports interesting data relating to the mucus Mu2 barrier changes during age advancement in mice. The manuscript is well written however several issues should be better addressed. As first English should be revised, and some sentences need to be clarified. For example, several markers analyzed change at a different time and should be discussed. Moreover, several data are confused maybe due to the low number of animals analyzed.

Major concerns

Animals: which diets have been used for animals? Is it the same for all treatment time? Please add diet characteristics and food intake consumption. Indicate if food and beverage were at libitinum. The number of animals for each group is very low, has been analyzed if they are enough for statistical analysis? If so indicate which method has been used.

Results:

Lines 195-196: this sentence is not clear, “diameter of mice…”

lines 174-177. Thickness  of the mucus layer was increased at 12 and 24 whereas at 18 was increased compared to 2 months but significantly reduced compared to 12 and 24 m. please discuss.

Fig 2B : reduction of Feret’s min diameter is lower in 12m mice compared to 18 and 24 mo. Please these results should be discussed

Fig 4B and C: results of fluorescence intensity analysis of MUC-2 indicate that only at 24mo there was a reduction however in gene expression analysis the authors report that Muc2 increased at 12mo  but no significant differences were reported for 2, 18, and 24 mo. This not correspondence should be discussed.

Lines 247-249: this sentence is not corrected considering that in Fig 5B is reported that at 24mo the number of globelet cells is significantly reduced compared to 2, 12, and 18 mo.

Fig 5E: why the expression of Klf4 is higher in 12 mo? please discuss.

Lines 274-276: this sentence should be revised, the expression of C1galt2 is shown in Fig6B, and the results reported in the figure indicate a higher expression at 12 mo. Please discuss.

Discussion:

The discussion is not well addressed to obtained result and it should be revised as also indicated in my suggestions in the results section.

Minor concerns:

Abstract :

please add gene analyzed

line 104: do they mean pH of colon content?

line 120: indicate which secondary antibody has been used

line 141: which blocking solution?

Reviewer 3 Report

An interesting study from Sang et al. was brought to this reviewer’s attention. The authors evaluated the mucus barrier and muc2-mucin expression in the colon tissue of experimental mice at different ages, including 2, 12, 18, and 24 months (n=3 mice/per month). This a timing and important work. However, major revision is needed to reach the publication level. The abstract needs review to explain better the methodological strategies used (e.g, how the mucus hole structure was investigated). The introduction is adequate. It would be important to clarify which cells produce C2Gnt2. That information is not clear in the introduction. Careful use and spelling of abbreviations are needed (use them immediately after the name is spelled). How do the authors deal with a small number of mice per group, especially when intragroup variations may happen?

The authors mentioned that colon fecal contents were removed, but in Fig. 1B, fecal/chow diet components seem to be found in the colonic lumen of mice. This could be improved by increasing the number of mice per group and avoiding this possible artifact, influencing mucosal (not mucus) layer thickness.

Authors need to add the info and version of the PS software in the methods. A negative control was used for the immunostaining; if so, add that information. Figure 1 is very interesting, however, it requires further clarification. For this reviewer, the mucus thickness (which can be adequately seen in Fig1A with muc2 immunostaining) seems clearly thicker in 12-mo old mice and significantly reduced in 24-month-old mice. This observation seems to be corroborated by Fig1B as well. In addition, bacterial invasion seems more evident in 24-mo old mice and perhaps in 18-mo old mice…why the authors did not quantify the intensity of EUB338 immunostaining?  Although legends are ok for the figures, it is important to describe the letters used to compare statistical significance between groups. Is it possible to use the Cryo-SEM technique to rotate images from Fig.2A to look for mucus layer thickness? Interestingly, greater DAPI immunolabeling can be seen with aging, which suggests higher cellularity with the interstitial lamina propria surrounding the colonic crypts (Fig. 4). Could the authors comment on reduced muc-2 expression density in the basis of the crypts? How could this affect stem cell activity? There is a contradictory finding of increased mucus layer with reduced muc-2 positive cells and immunolabeling in 24-mo-old mice…Is that right? In figure 5, it seems that the number of goblet cells is even more reduced at 24 months than at 2 months, and this is not shown in the graph (Fig. 5B). It would be interesting to comment on the role of polyamines on galactosylation of mucal glycoproteins in the discussion section.

Minor changes

line 309, needs ref after 16-month-old mice.

Correct liquid nitro-gen on line 101.

Round 2

Reviewer 1 Report

Thank you for the detailed response to my questions. I have no further comments. 

Reviewer 2 Report

After the changes done by the authors following the reviewers suggestions the manuscript resulted improved.